**REPORT**

# Stress-induced nucleolar rejuvenation via chaperone-mediated segregation in a filamentous fungus

Audra M. Rogers[2]📷, Brenna K. Broussard[2]📷, Rachel Taylor[1,2]📷, and Martin J. Egan[1,2]📷

**How the nucleolus recovers from acute proteostatic stress, particularly in multinucleate syncytia, remains poorly understood. In the highly polarized hyphae of the model filamentous fungus *Magnaporthe oryzae*, we uncover a novel stress-induced spatial quality control pathway that promotes the inheritance of rejuvenated nucleolar material during nuclear division. This pathway discriminates between newly formed and damaged nucleolar compartments, selectively partitioning and sequestering the latter. Our findings reveal a previously unrecognized mechanism for chaperone-mediated segregation of a membraneless nuclear organelle, extending principles of protein quality control to the unique demands of highly polarized syncytia.**

## Introduction

The nucleolus is a membraneless nuclear organelle that, in addition to ribosome biogenesis, contributes to the maintenance of nuclear proteostasis (Frottin et al., 2019; Mediani et al., 2019). Its multilayered architecture emerges through liquid–liquid phase separation driven by multivalent interactions among ribosomal RNA, scaffold proteins, and assembly factors (Feric et al., 2016; Lafontaine et al., 2021). Beyond its biosynthetic role, the nucleolus integrates growth and metabolic cues and undergoes rapid compositional and structural remodeling in response to cellular stress (Wang et al., 2019; Brunello et al., 2025).

During acute proteotoxic stress, misfolded proteins accumulate in the cytoplasm, nucleus, and within other organelles. Eukaryotic cells deploy spatial protein quality control systems that actively triage misfolded species to specific deposition sites according to their solubility, ubiquitination status, and refolding potential (Miller et al., 2015; Kaganovich et al., 2008). Most mechanistic insight into these deposition sites comes from budding yeast (*Saccharomyces cerevisiae*), where ubiquitinated, refoldable nuclear proteins are concentrated into the intranuclear quality control compartment (INQ) adjacent to the nucleolus (Miller et al., 2015), although comparable sequestration mechanisms have been reported in mammalian cells (Frottin et al., 2019; Mediani et al., 2019; Latonen, 2019). In the cytoplasm, soluble, ubiquitinated proteins are targeted to the juxtanuclear quality control compartment, whereas terminally aggregated or amyloid-like proteins are directed to the insoluble protein deposit near the vacuole (Escusa-Toret et al., 2013; Kaganovich et al., 2008). Importantly,

the nucleolus itself can also transiently detain misfolded nuclear proteins during stress, functioning as a reversible sequestration site that buffers nuclear homeostasis (Frottin et al., 2019; Mediani et al., 2019; Latonen, 2019). Prolonged stress can shift nucleolar material properties from liquid-like to gel- or solid-like states, leading to the formation of stable inclusions that reduce nucleolar dynamics and impair function (Audas et al., 2012; Gallardo et al., 2020). Clearance of these inclusions requires ATP-dependent chaperone systems, including Hsp70 and the disaggregase Hsp104 in yeast (Frottin et al., 2019; Gallardo et al., 2020). Thus, while the nucleolus contributes to maintaining nuclear proteostasis, it can also undergo stress-induced alterations, and how its normal architecture is re-established during recovery remains unclear.

Although the asymmetric inheritance of cytoplasmic protein aggregates to promote cellular rejuvenation is well established (Aguilaniu et al., 2003; Kaganovich et al., 2008; Spokoini et al., 2012), whether similar principles govern the partitioning of nuclear condensates such as the nucleolus remains unclear. The mechanisms by which a stressed nucleolus is rejuvenated prior to nuclear division, especially in large, multinucleate syncytia where protein quality control must operate across spatially complex domains, are largely unexplored. In the polarized hyphae of *Magnaporthe oryzae*, a model filamentous fungus and destructive crop pathogen, we identify a stress-induced spatial quality control pathway that promotes the preferential inheritance of rejuvenated nucleolar material during mitosis. This

[1]Department of Natural Sciences, Merrimack College, North Andover, MA, USA; [2]Department of Entomology and Plant Pathology, University of Arkansas, Fayetteville, AR, USA.

Correspondence to Martin J. Egan: eganma@merrimack.edu.

pathway discriminates between rejuvenated and damaged compartments, selectively partitioning and sequestering the latter. Our findings reveal a previously unrecognized mechanism for chaperone-mediated segregation of a membraneless nuclear organelle.

## Results and discussion

### Hsp104 enters the nucleus at the onset of mitosis following heat shock

We initially set out to examine how stress-induced cytoplasmic protein aggregation impacts nuclear dynamics in the polarized hyphae of *M. oryzae*. In filamentous fungi, rapid hyphal tip extension is tightly coupled to nuclear migration and cell cycle progression, and nuclei must migrate with the growing apex to sustain normal growth (Xiang, 2018). Exposure of *M. oryzae* hyphae to transient heat stress (42°C for 45 min) caused an immediate halt in both polarized growth and nuclear movement, which extended for hours into recovery (Fig. 1 A). During this arrest, Hsp104-GFP localized to numerous cytoplasmic foci, marking misfolded protein aggregates (Egan et al., 2015; Rogers and Egan, 2020) (Fig. 1 A and Video 1). After a period of recovery, growth polarity and nuclear dynamics gradually resumed. Notably, the arrested nucleus eventually underwent division, which coincided with the resumption of hyphal tip growth, linking cell cycle reentry to renewed morphogenesis. Strikingly, at the onset of mitosis, we observed a sudden accumulation of Hsp104-GFP within the nucleus, which also localized to a conspicuous ring-like structure adjacent to the chromatin (Fig. 1 B and Video 1). During mitotic progression, this Hsp104-enriched structure gradually disassembled. Interestingly, this process did not appear to require Hsp104's disaggregase activity, as Hsp104-labeled structures still diminish from the cytoplasm in Hsp104$^{DWB}$ mutants, which can bind aggregates but cannot refold or dissolve them (Hodson et al., 2012; Rogers and Egan, 2020) (Fig. 1 C and Video 2). Thus, Hsp104's ATP-dependent remodeling activity is dispensable for the removal of this structure, although Hsp104 may still facilitate its downstream disassembly, for example, by scaffolding access of other protein quality control (PQC) factors. *M. oryzae* undergoes a semi-closed mitosis, in which the nuclear envelope partially disassembles (Pfeifer and Khang, 2018, 2020), allowing nucleoplasmic proteins to leak into the cytoplasm at mitotic onset (Souza et al., 2004). To test whether Hsp104 nuclear entry was coupled to this event, we constructed a strain expressing Cet1-RFP together with Hsp104-GFP. Cet1 is an RNA 5′-triphosphatase and mRNA-capping enzyme that is localized in the nucleus during interphase and rapidly disperses upon nuclear envelope breakdown, allowing us to track compartment integrity in live cells. Imaging of heat-shocked hyphae coexpressing Cet1-RFP and Hsp104-GFP confirmed that Hsp104's nuclear entry is tightly coordinated with nuclear envelope breakdown at the onset of mitosis. As soon as Cet1-RFP began diffusing out of the nucleus, Hsp104-GFP influxed into the nucleus (Fig. 1, D and E), supporting a model in which Hsp104 is normally excluded from the nucleus but can enter when the nuclear permeability barrier relaxes at mitosis. Intriguingly, a similar regulated nuclear entry of Hsp104 has been observed in

quiescent yeast cells during stationary phase. In this system, as translation rates drop, Hsp104 is actively redirected to the nucleus to deal with accumulated misfolded proteins, and blocking Hsp104's nuclear import impairs recovery from dormancy (Kohler et al., 2024). Thus, Hsp104's partitioning between cytoplasm and nucleus can be dynamically controlled by both cell cycle and metabolic cues, ensuring it is deployed to safeguard nuclear proteostasis when needed.

### The nucleolus undergoes dynamic remodeling after heat stress

We hypothesized that this nuclear-localized structure represented the *M. oryzae* nucleolus. In support of this idea, *Aspergillus nidulans* (a model ascomycete) undergoes a "semi-closed" mitosis in which the nucleolus is extruded into the cytoplasm and disassembled during each division (Ukil et al., 2009). To directly test this idea, we generated a strain in which the nucleolar protein Nop1 (fibrillarin) was genetically tagged with RFP and imaged nucleolar dynamics in hyphae under control and heat-shocked conditions. In non-stressed hyphae, Nop1-RFP formed a compact nuclear subdomain adjacent to chromatin, consistent with the nucleolus (Fig. 2 A). During mitosis, this nucleolar signal dispersed at the onset of chromosome condensation and subsequently reformed within each daughter nucleus, ~11 min later (Fig. 2 A). Strikingly, however, in hyphae recovering from heat shock, we discovered that the nucleolus underwent a dramatic remodeling event prior to the onset of mitosis (Fig. 2, B and C). As hyphae recovered postheat stress, the original nucleolar compartment underwent a directed budding event, giving rise to a second nucleolar-like structure that remained contiguous with the original nucleolus (Fig. 2, B and C), producing a bilobed nucleolus. While phase separation likely contributes to directional nucleolar budding, we speculate that additional inputs such as chromatin anchoring or posttranslational modifications may be required for directional deformation and bud formation. When the nucleus entered mitosis, the newer nucleolar compartment behaved analogously to a normal nucleolus, disassembling at prophase for inheritance by daughter nuclei. In contrast, the "old" nucleolus, presumably harboring proteostatic damage from the heat shock, was expelled into the cytoplasm as nuclear division proceeded (Fig. 2 B). These observations suggest that the heat-damaged nucleolus is physically segregated, and its contents excluded, at least initially, from inheritance by daughter nuclei (Fig. 2 B).

To resolve the spatial and temporal fate of a core nucleolar protein during stress-induced remodeling, we generated a strain expressing Nop1 fused to the photoconvertible fluorescent protein mEos3.2. Nop1-mEos fluoresces green but can be irreversibly switched to a red-fluorescent state by violet light, allowing us to monitor the original nucleolar pool of Nop1 and track its behavior relative to newly synthesized Nop1 during remodeling (Zhang et al., 2012). We heat-shocked hyphae expressing Nop1-mEos, then immediately photoconverted the entire nucleolus (turning preexisting Nop1 "red") and monitored nucleolar behavior during recovery (Fig. 3, A–C). Initial buds were largely devoid of newly synthesized Nop1, indicating that the nascent nucleolar compartment was seeded, at least in part, by redistribution of existing (photoconverted) Nop1 from the old

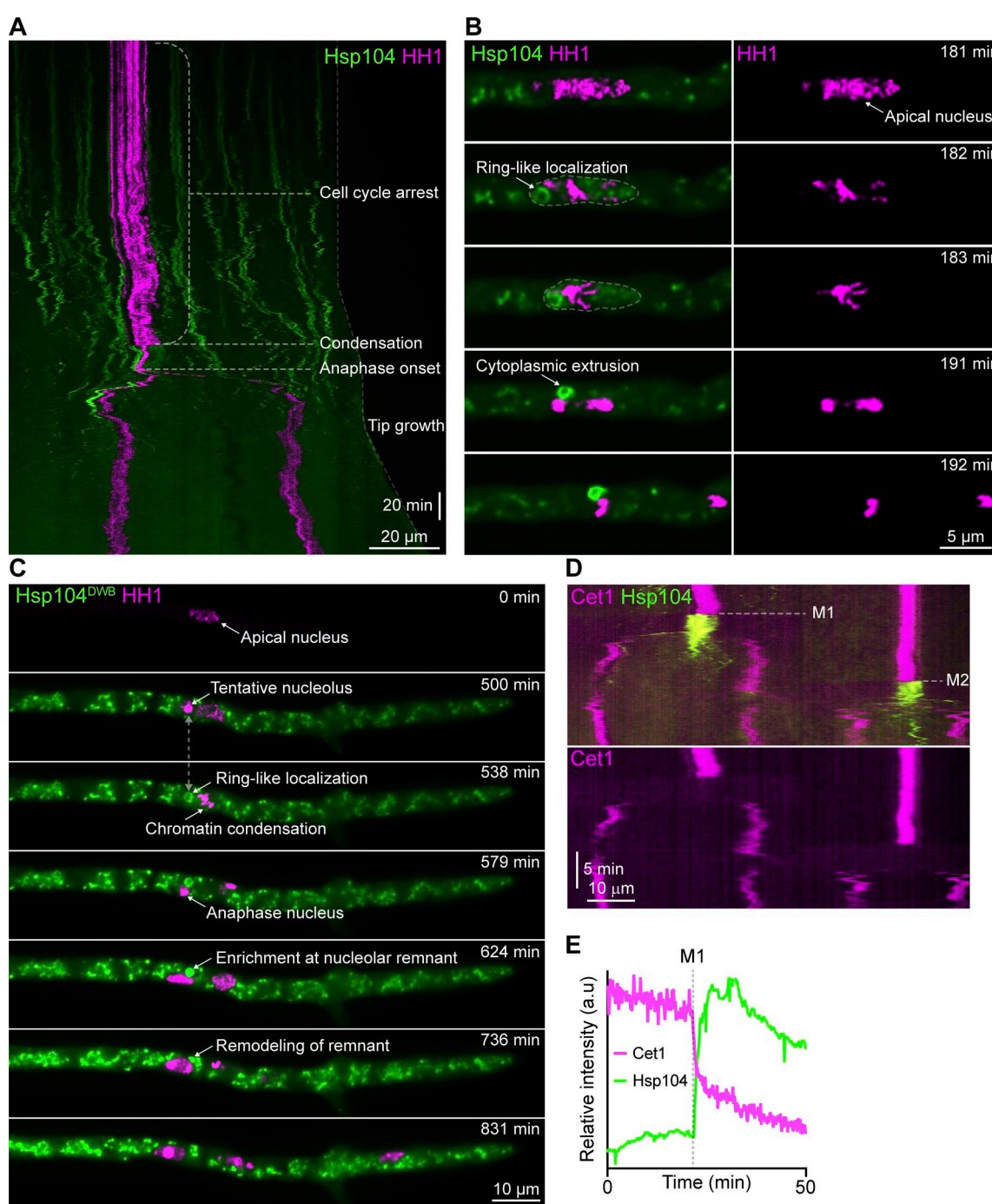

Figure 1. **Hsp104 enters the nucleus at mitotic onset following heat shock recovery. (A)** Kymograph from a time-lapse sequence showing nuclear (histone H1-RFP) and Hsp104-GFP dynamics in a single hypha recovering from a 45-min heat shock at 42°C. **(B)** Corresponding fluorescence micrographs from the same time-lapse sequence. Images were acquired every 10 s. **(C)** Time-lapse micrographs of a hypha expressing Hsp104$^{DWB}$-GFP and histone H1-RFP. **(D)** Kymographs from a time-lapse sequence showing Cet1-RFP and Hsp104-GFP dynamics in a single hypha. Upper panel shows merged channels; lower panel shows Hsp104-GFP alone. **(E)** Plot of relative fluorescence intensity over time for Hsp104-GFP (green) and Cet1-RFP (magenta) within a nuclear ROI in the same hypha ($n = 1$ nucleus). M1 indicates the onset of mitosis.

nucleolus into the bud (Fig. 3 D). As the bud grew larger, however, green fluorescence in the new compartment increased relative to the old compartment, suggesting incorporation of newly synthesized Nop1 into the maturing structure (Fig. 3 E). We speculate that this phase may coincide with the re-establishment of rDNA chromatin interactions around the newly formed

nucleolar compartment, a process that requires active rRNA transcription and may correspond to the onset of new Nop1 synthesis (Németh and Grummt, 2018). During mitosis, the photoconversion marked clearly that the new nucleolus disassembled and its constituents (including both old and new Nop1) partitioned into the two nascent daughter nucleoli,

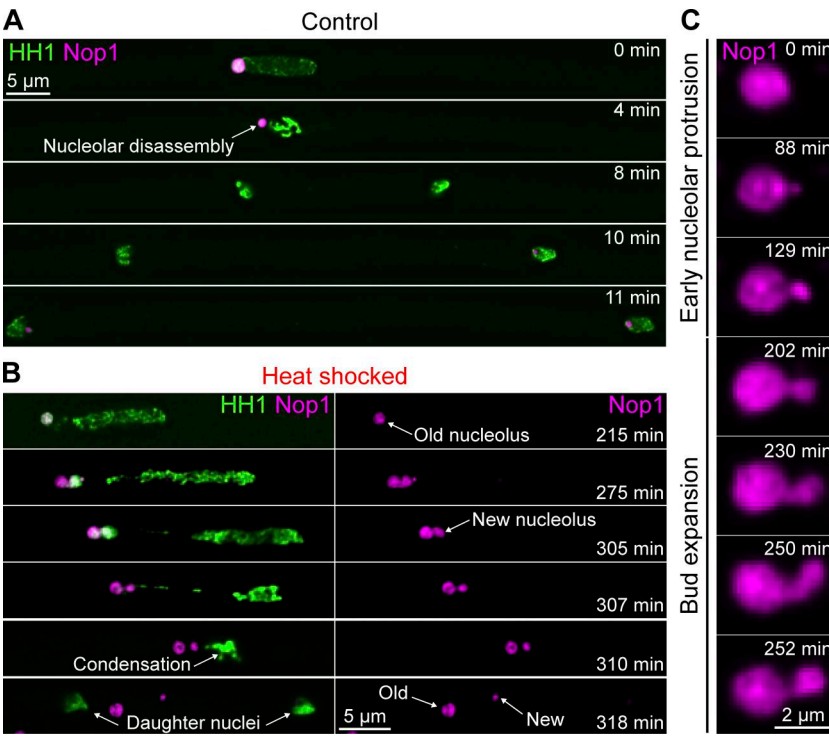

**Figure 2. Heat shock induces remodeling of the nucleolus prior to mitotic onset. (A)** Time-lapse micrographs of a hypha co-expressing Nop1-RFP and histone H1-GFP under control conditions, showing typical nucleolar disassembly and segregation during nuclear division. **(B)** Corresponding time-lapse micrographs of a heat-shocked hypha showing nucleolar remodeling prior to mitotic onset. Hyphae were heat shocked at 42°C for 45 min before imaging. **(C)** Time-lapse micrographs of a heat-shocked hypha expressing Nop1-RFP, showing apparent nucleolar budding and expansion during recovery. Images were acquired every 10 s; selected time points post heat shock are shown.

whereas the bulk of the old nucleolar remnant was extruded and eventually diminished in the cytoplasm (Fig. 3 F and Video 3). Critically, nucleolar remodeling was only observed for the first mitosis following heat stress and subsequent nuclear divisions in the same hypha proceeded normally with a single nucleolus that disassembled/reformed within daughter nuclei (Video 3). Thus, nucleolar remodeling and remnant extrusion represent a one-time, stress-triggered response to acute proteotoxic insult, presumably to eliminate persistent damage accumulated during heat shock. Once the damaged compartment diminishes, subsequent divisions proceed without remodeling, likely because newly assembled nucleoli are largely free of misfolded proteins. This selective reuse of existing nucleolar material is reminiscent of templated organelle biogenesis in other systems, where inheritance from preexisting structures accelerates recovery and maintains architectural fidelity compared with de novo assembly (Lowe and Barr, 2007; Warren and Wickner, 1996; Rossanese et al., 2001; Loncarek and Bettencourt-Dias, 2018). Such templating may be particularly advantageous after acute stress, enabling rapid formation of a rejuvenated nucleolus while physically excluding damaged material.

## Hsp104 segregates the old from the new nucleolar compartment

We examined Hsp104's involvement in this nucleolar quality control pathway by constructing a strain co-expressing Hsp104–GFP and Nop1-RFP. Consistent with our earlier observations, Nop1–RFP in this strain also showed consistent nucleolar budding and bilobe formation after heat shock. Remarkably, at mitotic onset, Hsp104–GFP localized specifically to the old nucleolar compartment (Fig. 4, A–C and Video 3), demarcating the

boundary between old and "new" nucleolar material within a seemingly contiguous, bilobed structure (Fig. 4, A and B). The selective recruitment of Hsp104 to the old nucleolar remnant suggests that it is recognized as a distinct quality control substrate during mitotic entry. In yeast, Hsp104 targeting to protein inclusions can be mediated by small heat shock proteins such as Hsp42 or co-chaperones, including Sis1 and Btn2, which bind misfolded proteins and anchor them to deposition sites (Specht et al., 2011; Malinovska et al., 2012; Miller et al., 2015). Posttranslational modifications, such as ubiquitination and SUMOylation of nucleolar proteins, have been implicated in marking substrates for sequestration into quality control compartments (Gallina et al., 2015; Latonen, 2019). Such recognition may also be influenced by stress-induced changes in the material properties of the old nucleolus, as damaged condensates often transition from a dynamic, liquid-like state to a more rigid, gel-like or solid state, which can alter protein mobility and chaperone accessibility (Riback et al., 2017; Frottin et al., 2019; Mediani et al., 2019). Together, these mechanisms could enable Hsp104 or its cofactors to selectively recognize the damaged nucleolus, distinguishing it from the newly assembled replacement.

As mitosis proceeded and the new nucleolus disassembled, the old nucleolar mass, now marked intensely by Hsp104–GFP, was expelled into the cytoplasm. Hsp104 remained associated with this cytoplasmic nucleolar remnant, which gradually diminished in size and eventually disappeared, consistent with downstream processing, although the specific cytoplasmic pathways involved remain to be determined (Fig. 4 C; and Videos 4, 5, and 6). Interestingly, in some instances, Hsp104 appears to access the old nucleolar compartment prior to widespread nuclear entry at envelope breakdown, suggesting an earlier and more regulated

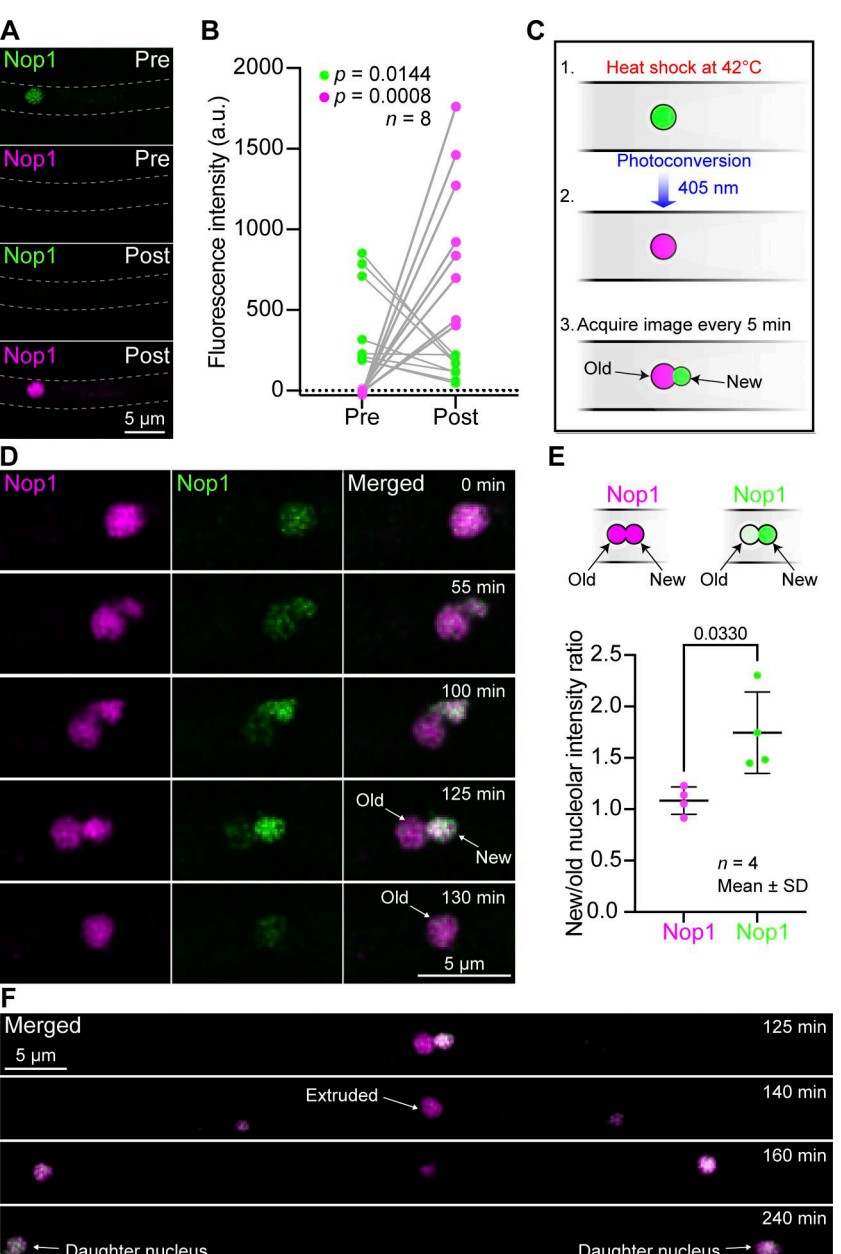

Figure 3. **Nucleolar budding initiates from preexisting material but incorporates newly synthesized Nop1 during expansion. (A)** Fluorescence micrographs of a hypha expressing Nop1-mEos before ("Pre") and after ("Post") photoconversion, shown as green (non-converted) and magenta (photoconverted) channels. Dashed lines indicate the outline of the hypha. **(B)** Quantification of Nop1-mEos fluorescence intensity before and after photoconversion. Green (non-converted) and magenta (photoconverted) signals were measured within the same nucleolar ROI in individual nuclei. Each pair of points represents measurements from a single nucleolus before and after photoconversion. P values (P = 0.0144 for the green channel and P = 0.0008 for the magenta channel) were calculated using two-tailed paired $t$ tests comparing pre- and postphotoconversion fluorescence intensity for each channel independently. Statistical testing was used to confirm the consistency and magnitude of photoconversion across nucleoli. **(C)** Schematic outlining the experimental workflow: cells were heat shocked, photoconverted, and imaged every 5 min in both green and magenta channels during nucleolar remodeling and mitosis. **(D)** Time-lapse fluorescence micrographs of a hypha expressing Nop1-mEos during recovery from heat shock, showing remodeling of a single nucleolus over time. Photoconverted (magenta) and newly synthesized (green) Nop1 are shown separately and merged. A distinct nucleolar bud emerges from preexisting material and progressively incorporates new Nop1. **(E)** Quantification of Nop1-mEos intensity in the newly formed new versus preexisting old nucleolar compartments shown in D. Fluorescence ratios were calculated separately for photoconverted (magenta) and newly synthesized (green) channels. Each point represents a single nucleolus; P value (P = 0.0330) was calculated using a two-tailed unpaired $t$ test. Data are presented as mean ± SD ($n$ = 3 nucleoli). **(F)** Continuation of the time-lapse sequence shown in D, highlighting the fate of the old (photoconverted) nucleolus.

mechanism of chaperone access (Fig. 4 C). Across 35 apical nuclei examined from 10 biological replicates (2–5 hyphae per replicate), 23 (~66%) showed clear nucleolar budding, selective Hsp104 recruitment, and extrusion of the preexisting nucleolar compartment, indicating that this behavior is a frequent outcome of heat shock recovery.

SUMO conjugation is a conserved and highly stress-responsive posttranslational modification that predominantly targets nuclear proteins, modulating their interactions, activity, and localization and thereby reshaping nuclear proteostasis during stress responses (Flotho and Melchior, 2013; Cubeñas-Potts and Matunis, 2013). In some contexts, poly-SUMOylated substrates are recognized by SUMO-targeted ubiquitin ligases, functionally linking SUMO signaling to ubiquitin-mediated remodeling, sequestration, or regulated turnover of nuclear proteins (Kumar et al., 2022; Keiten-Schmitz et al., 2020). To assess whether the

preexisting nucleolar compartment contains stress-modified substrates, we examined SUMO dynamics using a GFP–SUMO reporter. Following heat shock, SUMO rapidly accumulated throughout the nucleus (Fig. S1 A), consistent with activation of nuclear protein quality control pathways. As cells approached mitosis, SUMO became selectively enriched at a single nuclear focus whose timing and position were consistent with the preexisting (old) nucleolar compartment identified in Nop1-labeled strains (Fig. S1, B and C; and Videos 7 and 8). The SUMO-enriched structure persisted transiently in the cytoplasm after extrusion (Video 8). These observations suggest that the extruded nuclear remnant is enriched in SUMO-modified material, consistent with selective recognition and sequestration of the preexisting nucleolar compartment during recovery.

Finally, we asked whether additional chaperones participate in recognizing the preexisting nucleolar compartment. In a

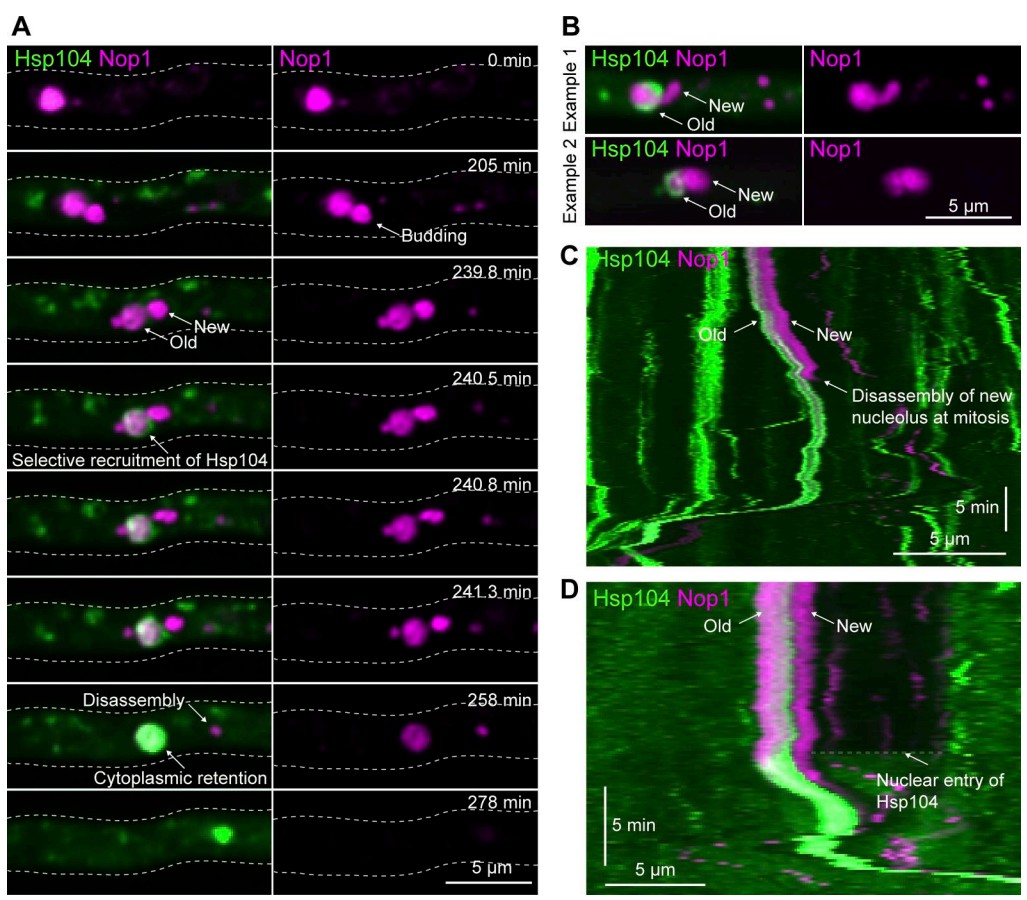

Figure 4. **Hsp104 selectively associates with the preexisting nucleolar compartment before mitotic onset. (A)** Time-lapse fluorescence micrographs of a hypha co-expressing Hsp104-GFP and Nop1-RFP during recovery from heat shock, showing nucleolar budding and selective recruitment of Hsp104 to the preexisting compartment prior to mitosis. **(B)** Additional examples showing preferential Hsp104 association with the preexisting nucleolar compartment. **(C and D)** Kymographs from time-lapse sequences of hyphae co-expressing Hsp104-GFP and Nop1-RFP, showing selective Hsp104 recruitment to the pre-existing compartment prior to the onset of mitosis.

strain co-expressing Hsp70–RFP and Hsp104–GFP, the two chaperones co-localized at cytoplasmic stress-induced aggregates and entered the nucleus together at mitotic onset. Both chaperones then became enriched at a single nuclear focus at the expected position of the preexisting nucleolus and remained associated with the extruded remnant (Video 9). These observations suggest that multiple PQC factors, including Hsp70 and Hsp104, recognize the old nucleolar compartment prior to its segregation, consistent with a conserved role for chaperone networks in managing stress-modified nuclear material (Miller et al., 2015).

Whether the extruded nucleolus is truly damaged or simply serves as a sequestration site remains unresolved. The close parallels to the INQ in yeast, a nucleolus-associated site for se-questration of ubiquitinated, misfolded nuclear proteins during stress that can be evicted from the nucleus under certain conditions (Miller et al., 2015; Kumar et al., 2022), raise the possibility that the old nucleolus represents a related form of stress-induced disposal of a nuclear compartment. If so, extrusion may spatially isolate aggregation-prone components from daughter nuclei, contributing to mitotic rejuvenation. In a syncytial cytoplasm, complete removal of damaged nucleolar remnants may be

particularly critical, as persistent misfolded material could other-wise be reincorporated by other migrating nuclei during subsequent divisions. By sequestering or excluding these remnants, spatial quality control safeguards nuclear integrity across the shared cytoplasm (Jalihal and Gladfelter, 2025).

The parallels between stress-induced nucleolar remodeling in *M. oryzae* and nuclear PQC pathways described in budding yeast and mammalian cells suggest that these phenomena reflect conserved organizational principles. In yeast, the INQ forms adjacent to the nucleolus and is enriched for ubiquitinated, chaperone-bound nuclear proteins during stress, with SU-MOylation contributing to substrate selection and compartment dynamics (Kumar et al., 2022). In mammalian cells, the nucleolus likewise functions as a stress-responsive condensate that transiently sequesters nuclear proteins and undergoes pronounced structural remodeling under proteotoxic conditions (Frottin et al., 2019). Together with our observations that the old nucleolar compartment in *M. oryzae* becomes enriched in SUMO-modified material and recruits conserved chaperones such as Hsp70 and Hsp104, these findings support the idea that spatial segregation of stress-modified nuclear material represents an evolutionarily conserved strategy for preserving nuclear

function during recovery. While the mechanisms differ across systems, the shared involvement of chaperones and SUMOylation suggests broad conservation of this nuclear quality control strategy.

In summary, our findings reveal a previously unrecognized stress-inducible pathway for quality control of a membraneless nuclear organelle in a model filamentous fungus and major crop pathogen (Wilson and Talbot, 2009) (Fig. 5). Following acute proteotoxic stress, the nucleolus remodels into two physically linked compartments. One is a newly assembled structure that we speculate is largely free of damage and destined for inheritance. The other is an older remnant that accumulates SUMO-modified material, recruits the chaperones Hsp70 and Hsp104, and is subsequently extruded into the cytoplasm. We propose that this selective segregation limits transmission of damaged nucleolar components to daughter nuclei and safeguards nuclear proteostasis during recovery. This mechanism extends principles of asymmetric damage segregation that have been established for organelle-associated cytoplasmic inclusions and for non-syncytial cells in which aggregates are differentially partitioned at mitosis (Sontag et al., 2017; Spokoini et al., 2012; Ogrodnik et al., 2014) to a phase-separated nuclear organelle in a syncytial system. More broadly, syncytial and multinucleate cells like filamentous fungi likely rely on spatial quality control to maintain nuclear integrity across shared cytoplasm.

## Materials and methods

### Fungal culture and transformation

*M. oryzae* strains were cultured on complete media (CM) containing glucose (10 g L$^{-1}$), peptone (2 g L$^{-1}$), yeast extract (1 g L$^{-1}$), casamino acids (1 g L$^{-1}$), NaNO$_3$ (6 g L$^{-1}$), KCl (0.5 g L$^{-1}$), MgSO$_4$ (0.5 g L$^{-1}$), and KH$_2$PO$_4$ (1.5 g L$^{-1}$), supplemented with 0.1% (vol/vol) vitamin solution and 0.1% (vol/vol) trace element solution. Medium pH was adjusted to 6.5 with NaOH prior to sterilization. For solid medium, agar was added to 15 g L$^{-1}$. Media were autoclaved at 121°C for 20 min, cooled to ~50°C, supplemented with penicillin–streptomycin (50 U ml$^{-1}$), poured into 6-cm Petri dishes, and stored at 4°C for up to 4 wk (Molinari and Talbot, 2022). Vegetative plate cultures were maintained on CM at 25°C with a 12-h light/12-h dark cycle for 7–12 days. Protoplasts of the *M. oryzae* entry strain were generated from mycelium grown on CM plates. Briefly, mycelium was excised, homogenized in complete liquid medium, and incubated at 23°C with shaking (120 rpm) for 48 h. Mycelial biomass was harvested by filtration and digested with VinoTaste Pro (Novozymes) in OM buffer (1.2 M MgSO$_4$ and 10 mM Na$_2$HPO$_4$/NaH$_2$PO$_4$, pH 5.8) to generate protoplasts. Protoplasts were purified by density separation using ST buffer (0.6 M sorbitol and 0.1 M Tris-HCl, pH 7.0), washed in STC buffer (1.2 M sorbitol, 10 mM Tris-HCl, pH 7.5, and 10 mM CaCl$_2$), and used for polyethylene glycol–mediated transformation with >2 μg DNA per 5 × 10$^6$ protoplasts (Talbot et al., 1993). Strains expressing Hsp104-GFP and HSP104$^{DWB}$-GFP (Rogers and Egan, 2020) were transformed with Hsp70-RFP, Cet1-RFP, Nop1-RFP, Nop1-mEos3.2, or a histone H1-RFP (Saunders et al., 2010) constructs, and transformants were selected for resistance to phosphinothricin, conferred by a bar cassette on pCB1530 or pCB1531 (Sweigard et al., 1997; Carroll

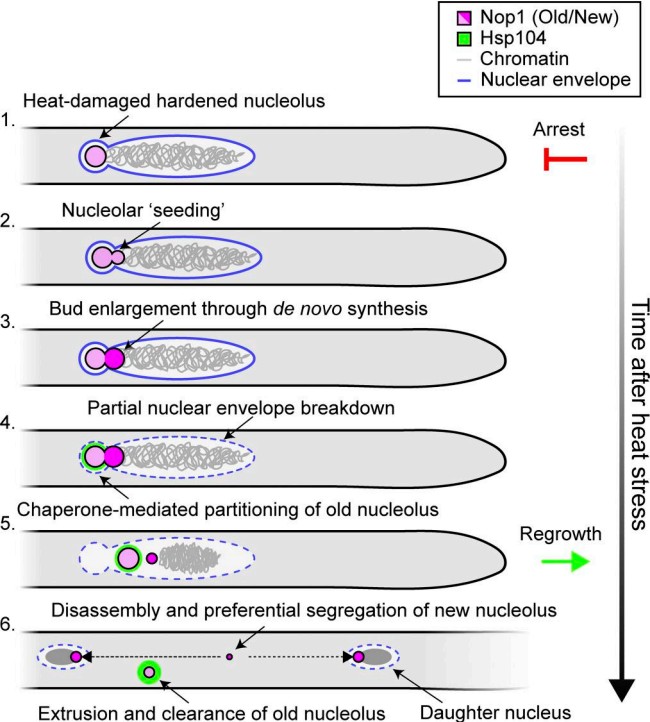

Figure 5. **Model for nucleolar remodeling, partitioning, and quality control following heat shock.** Schematic illustration of nucleolar remodeling in *M. oryzae* during recovery from heat stress. (1) Heat shock damages the existing nucleolus. (2–3) A new nucleolar bud emerges from the old (preexisting) nucleolus and expands through de novo synthesis. (4) Partial nuclear envelope breakdown permits entry of the molecular chaperones such as Hsp104 and Hsp70. (5) The old nucleolar compartment accumulates SUMO-modified material and selectively recruits Hsp70 and Hsp104, mediating the partitioning of old and new nucleolar material. (6) The newly formed nucleolus disassembles at mitotic onset and is preferentially inherited by daughter nuclei, while the old nucleolus is extruded and diminishes.

et al., 1994). A strain expressing HH1-RFP (Saunders et al., 2010) was transformed with a GFP-SUMO construct, and transformants were selected for resistance to hygromycin, conferred by a hyg cassette on pCB1004 (Carroll et al., 1994).

The Cet1-RFP:Hsp104-GFP strain was generated by transforming the Hsp104-GFP strain with a Cet1-RFP plasmid, with transformants selected for phosphinothricin resistance (Sweigard et al., 1997). Strains co-expressing Hsp70-RFP:Hsp104-GFP, Nop1-RFP:Hsp104-GFP, Nop1-RFP:Hsp104$^{DWB}$-GFP, histone H1-tdTom:Hsp104-GFP, and histone H1-tdTom:Hsp104$^{DWB}$-GFP were constructed the same way. A strain co-expressing histone H1-GFP and Nop1-RFP was generated by transforming the Nop1-RFP-ILV1 parent strain (chlorimuron ethyl-resistant), with histone H1-GFP plasmid, and transformants were selected for phosphinothricin resistance. All transformations were carried out using polyethylene glycol–mediated transformation of protoplasts, resulting in either homologous recombination–mediated gene modification or ectopic integration of gene fusion constructs, as previously described (Talbot et al., 1993).

### Plasmid construction

DNA constructs for fluorescent fusion proteins were assembled using either In-Fusion cloning (Clontech) or yeast gap repair

from linear PCR products amplified with high-fidelity Phusion polymerase (New England Biolabs). PCR primers were designed in SnapGene (version 4.3.10; GSL Biotech), and DNA sequences were retrieved from FungiDB (https://fungidb.org). A complete list of PCR primers used for cloning and strain construction is provided in Table S1.

Cet1-RFP and Nop1-RFP, and Hsp70-RFP constructs were generated by replacing the promoter and ORF of a Scad2–RFP–Bar plasmid, derived from Scad2–GFP–ILV (provided by the Talbot laboratory), with PCR amplicons of the respective gene. The Nop1–mEos3.2–Bar construct was assembled by replacing RFP with mEos3.2, which was amplified from a codon-optimized *M. oryzae* mEos3.2 sequence. This sequence was synthesized and cloned by Integrated DNA Technologies using blunt-end ligation. A flexible Gly–Ala$_5$ (GA$_5$) linker was added to the 5′ end of mEos3.2 using primers GA5.mEos3.2.F and GA5.mEos3.2.R. The GA$_5$–mEos3.2 PCR product was A-tailed with Taq polymerase and cloned into the pCR2.1-TOPO vector to produce pARmEos. A histone H1-GFP fusion construct was generated by replacing the tdTomato-coding sequence in the histone H1-tdTom construct with eGFP, which was then cloned into the pCB1531 backbone. The GFP-SUMO construct was generated by assembling three PCR-amplified fragments into a vector backbone linearized by PCR amplification of pCB1004. Two fragments corresponding to the SUMO genomic locus (the *M. oryzae* ortholog of yeast Smt3) were amplified from Guy11 genomic DNA. Fragment 1 encompassed ~2 kb upstream of the *SUMO* ORF and was used as the native *SUMO* promoter, while fragment 3 contained the *SUMO* ORF. A third fragment containing the GFP-coding sequence and associated regulatory elements was amplified from a previously assembled histone H1-GFP construct. These fragments were assembled to generate an N-terminal GFP fusion of the *SUMO* ORF.

### DNA isolation and extraction
Genomic DNA was extracted from mycelium harvested from 2-day-old liquid cultures of *M. oryzae* using a modified cetyltrimethylammonium bromide (CTAB) protocol. Mycelium grown in liquid culture and harvested by filtration was flash-frozen in liquid nitrogen and ground to a fine powder. Tissue was lysed in prewarmed CTAB buffer (65°C), extracted twice with chloroform:isoamyl alcohol, and DNA was precipitated with isopropanol. Following ethanol washing, DNA was resuspended in sterile water (Molinari and Talbot, 2022). PCR was performed with Taq polymerase (GenScript), and products were analyzed by standard agarose gel electrophoresis. PCR amplicons were purified using the Wizard SV Gel and PCR Clean-Up System (Promega), and plasmids were isolated from *Escherichia coli* liquid culture using the Wizard Plus SV Minipreps DNA Purification System (Promega).

### Image acquisition and analysis
Fluorescence microscopy was performed on a Nikon Ti-E Eclipse inverted microscope equipped with a 100×/1.49 NA oil-immersion Apo TIRF objective (Nikon), a Perfect Focus System (Nikon), and a motorized piezo stage (Nikon). Fluorescence emitted from eGFP and RFP-T was detected using a Zyla 4.2 sCMOS camera (Andor Technology). Fluorophores were excited at 405, 485, and 560 nm using an AURA II solid-state triggered illuminator (Lumencor). Images were viewed, and hardware was controlled using NIS-Elements AR software (version 4.60; Nikon). *M. oryzae* hyphae were imaged at room temperature (21–23°C). Time-lapse sequences were deconvolved with spherical aberration correction and background subtraction under the "automatic" setting for 2D datasets using NIS-Elements AR software. Image manipulations were performed in ImageJ2 software (version 2.16.0/1.54p); National Institutes of Health).

Fluorescence quantification was performed using FIJI (ImageJ; National Institutes of Health) on non-deconvolved raw time-lapse image stacks acquired as .nd2 files. Image sequences were imported using the Bio-Formats plugin to preserve metadata and acquisition parameters. For each time-lapse series, regions of interest (ROIs) were manually defined based on cellular morphology using the ROI Manager and applied consistently across all frames. Mean fluorescence intensity within each ROI was quantified across the time series using the "Plot Z Axis profile" function. Where appropriate, background fluorescence was determined from an adjacent, cell-free region and subtracted from all measurements to account for camera offset and diffuse background signal. Where applicable, fluorescence values were normalized to the initial frame or to the maximum intensity within the time series to allow comparison between cells and independent experiments.

Kymograph analysis was performed in FIJI to visualize spatial redistribution of fluorescence over time. A segmented line was drawn along the longitudinal axis of individual hyphae, extending from the subapical region to the tip. Kymographs were generated using the "Reslice" function, producing a 2D representation in which the x-axis corresponds to distance along the hypha and the y-axis represents time. Fluorescence intensity along the line was plotted across successive frames, enabling visualization and quantification of dynamic fluorescence displacement, accumulation, or transport along the hyphal axis. All kymographs were generated from the same line width and orientation to ensure consistency across samples.

### Sample preparation and treatment
Hyphae from the active edge of 2–3 day-old vegetative plate cultures were excised from CM plates and inverted onto the coverslip of a 35-mm FluoroDish cell culture dish (World Precision Instruments) for imaging. For heat shock, FluoroDishes were placed at the bottom of a 42°C hybridization oven for 45 min and imaged immediately. Recovery is designated as incubation at room temperature (22–24°C). Imaging was initiated immediately following removal from heat stress, with time stamps indicating elapsed minutes after sample removal.

### Photoconversion of Nop1-mEos3.2
Hyphae from a Nop1-mEos3.2–expressing strain were heat shocked as described above and returned to room temperature for imaging. The apical-most nucleus in hyphae at the leading edge of the colony was identified by Nop1-mEos3.2 fluorescence using 488 nm excitation. Photoconversion was performed with a 405 nm LED light source (AURA II, Lumencor) at 100% intensity for 1 min. Immediately after photoconversion, images were

acquired in both green and red channels for up to 16 h at 5-min intervals using the same imaging conditions described above.

### Statistical analysis and data presentation

All statistical analyses were performed in Prism 9 (GraphPad Software, version 9.4.1). Data are presented as mean ± SD unless otherwise stated. The specific statistical tests used are indicated in the corresponding figure legends. It was assumed that the data distribution is normal, but this was not formally tested. All plots were generated in Prism 9.

### Figure and video preparation

Figures were assembled in Adobe Illustrator (version 28.6), and videos were prepared in Adobe Premiere Pro (version 25.1.0).

### Online supplemental material

Live-cell fluorescence microscopy was used to monitor nucleolar dynamics, SUMO redistribution, and chaperone behavior during recovery from heat shock (42°C for 45 min). Heat shock induced global nuclear accumulation of SUMO, followed by focal enrichment at a position consistent with the old nucleolar compartment prior to mitotic entry (Fig. S1; and Videos 7 and 8). Time-lapse imaging revealed transient nuclear entry of Hsp104 at mitotic onset (Video 1) and preferential recruitment of Hsp104 to the old nucleolar compartment during nucleolar remodeling across independent biological replicates (Videos 4, 5, and 6). Nucleolar budding and extrusion events were observed exclusively during the first mitosis following heat shock (Video 3). Diminishment of extruded nucleolar remnants occurred even when Hsp104 disaggregase activity was impaired, indicating that remnant resolution can proceed independently of disaggregase activity (Video 2). Co-imaging of Hsp70 and Hsp104 demonstrated coordinated recruitment to the preexisting nucleolus and continued association with the extruded remnant during recovery (Video 9). Table S1 shows the primer list used in this study.

## Data availability

The data are available from the corresponding author upon reasonable request.

## Acknowledgments

We thank Baronger Beiger and Rinalda Proko (formerly University of Arkansas) for early contributions to this project and Nick Talbot and Lauren Ryder (The Sainsbury Lab, Norwich, UK) for sharing reagents. The authors used generative AI software (ChatGPT, OpenAI) to assist with language refinement of the manuscript. All scientific content and conclusions are the authors' own.

This work was supported in part by a National Institutes of Health Academic Research Enhancement Award (R15 GM132869) to M.J. Egan.

Author contributions: Audra M. Rogers: data curation, formal analysis, investigation, methodology, resources, validation, visualization, and writing—original draft, review, and editing. Brenna K. Broussard: formal analysis, investigation, validation, and writing—review and editing. Rachel Taylor: methodology. Martin J. Egan: conceptualization, data curation, formal analysis, funding acquisition, investigation, methodology, project administration, supervision, validation, visualization, and writing—original draft, review, and editing.

Disclosures: The authors declare no competing interests exist.

Submitted: 11 August 2025

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

# Supplemental material

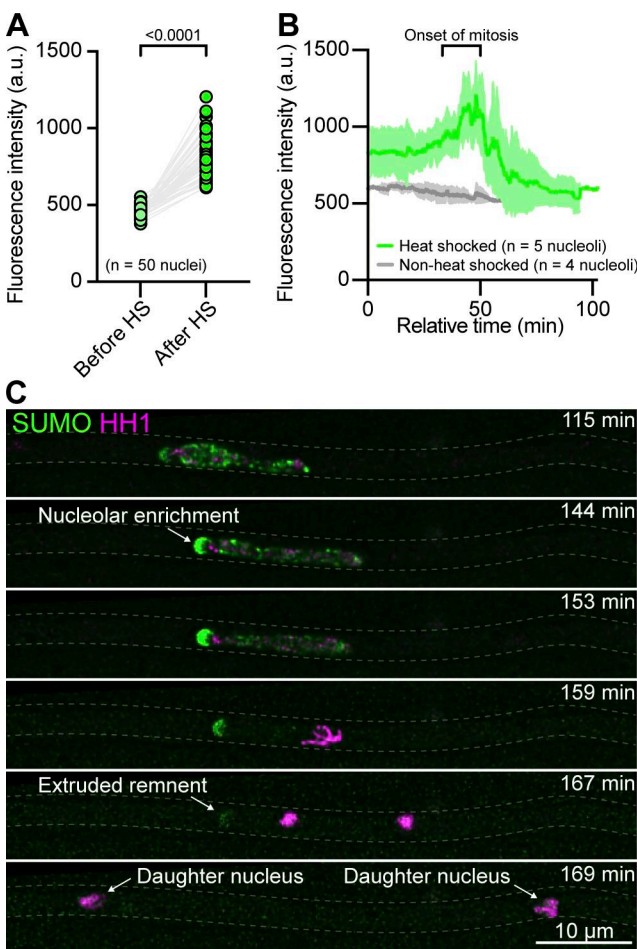

Figure S1.  **Heat shock induces nuclear SUMO accumulation and focal enrichment at a site consistent with the preexisting nucleolar compartment.** **(A)** Quantification of whole-nucleus GFP-SUMO fluorescence intensity before and after heat shock (42°C for 45 min). Each pair of points represents a single nucleus, with paired values connected by thin gray lines ($n$ = 50 nuclei). Data were analyzed using a two-tailed paired $t$ test (P < 0.0001). **(B)** Time-aligned GFP-SUMO fluorescence intensity traces from heat-shocked and non–heat-shocked hyphae, measured at a defined nuclear region corresponding to the nucleolar position. The green and gray lines represent the mean intensity for heat-shocked ($n$ = 5 nucleoli) and non–heat-shocked ($n$ = 4 nucleoli) cells, respectively; shaded regions indicate ±SD. The approximate timing of mitotic onset is indicated. **(C)** Representative time-lapse sequence of a heat-shocked hypha co-expressing GFP-SUMO (green) and histone H1-RFP (magenta), illustrating diffuse nuclear accumulation followed by focal SUMO enrichment prior to mitosis. This sequence corresponds to Video 7.

Video 1. **Hsp104 enters the nucleus at mitotic onset following heat shock recovery.** Time-lapse fluorescence video of a fungal hypha co-expressing Hsp104-GFP (green) and histone H1-RFP (magenta) during recovery from heat shock, showing nuclear entry of Hsp104 at mitotic onset. This video corresponds to the kymograph and micrographs shown in Fig. 1, A and B. Images were acquired every 10 s and displayed at 30 frames per second. The time stamp indicates minutes after the end of heat shock.

Video 2. **Nucleolar remnants still diminish in the absence of Hsp104 disaggregase activity.** Time-lapse fluorescence video of a hypha co-expressing Hsp104$^{DWB}$-GFP and histone H1-RFP during recovery from heat shock (42°C for 45 min). The extruded nucleolar remnant progressively diminishes over time in the Hsp104$^{DWB}$ mutant, despite the persistence of cytoplasmic aggregates, indicating that Hsp104's disaggregase activity is not strictly required for remnant processing. This video corresponds to the micrographs shown in Fig. 1 C. Images were acquired every 10 s and displayed at 60 frames per second. The time stamp indicates minutes after the end of heat shock.

Video 3. **Nucleolar remodeling and extrusion events are restricted to the first mitosis after heat shock.** Time-lapse fluorescence video of a hypha expressing Nop1-mEos (green = unconverted; magenta = converted) during recovery from heat shock. The video shows nucleolar budding and extrusion following photoconversion (corresponds to Fig. 3, D and F). A second mitosis occurs later in the same hypha, during which no budding or extrusion is observed. Images were acquired every 5 min and displayed at 10 frames per second. The time stamp indicates minutes after photoconversion.

Video 4. **Hsp104 is selectively recruited to the old nucleolar compartment during recovery from heat shock.** Time-lapse fluorescence video of hyphae co-expressing Hsp104-GFP (green) and Nop1-RFP (magenta) during recovery from heat shock (42°C for 45 min), showing nucleolar budding and preferential recruitment of Hsp104 to the preexisting (old) nucleolar compartment. At the time of recruitment, the old and new compartments remain physically connected. This video corresponds to the time-lapse sequence shown in Fig. 4 A and represents an independent biological replicate. Images were acquired every 10 s and displayed at 30 frames per second. The time stamp indicates minutes after the end of heat shock.

Video 5. **Second example of selective recruitment of Hsp104 to the old nucleolar compartment during recovery from heat shock.** Time-lapse fluorescence video of hyphae co-expressing Hsp104-GFP (green) and Nop1-RFP (magenta) during recovery from heat shock (42°C for 45 min), showing nucleolar budding and preferential recruitment of Hsp104 to the preexisting (old) nucleolar compartment. This video corresponds to the bottom panel of Fig. 4 B ("Example 2") and the kymograph shown in Fig. 4 C and represents an independent biological replicate. Images were acquired every 10 s and displayed at 30 frames per second. The time stamp indicates minutes after the end of heat shock.

Video 6. **Third example of selective recruitment of Hsp104 to the old nucleolar compartment during recovery from heat shock.** Time-lapse fluorescence video of hyphae co-expressing Hsp104-GFP (green) and Nop1–RFP (magenta) during recovery from heat shock (42°C for 45 min), showing nucleolar budding and preferential recruitment of Hsp104 to the preexisting (old) nucleolar compartment. This video corresponds to the top panel of Fig. 4 B ("Example 1") and the kymograph shown in Fig. 4 D and represents an independent biological replicate. Images were acquired every 10 s and displayed at 30 frames per second. The time stamp indicates minutes after the end of heat shock.

Video 7. **SUMO becomes enriched at a nuclear focus consistent with the preexisting nucleolar compartment prior to mitosis.** Time-lapse fluorescence video of hyphae co-expressing GFP-SUMO (green) and histone H1-RFP (magenta) during recovery from heat shock (42°C for 45 min). SUMO initially accumulates throughout the nucleus and then becomes selectively enriched at a single nuclear focus whose position and timing are consistent with the preexisting (old) nucleolar compartment identified in Nop1-labeled strains. This enrichment occurs before mitotic onset. This video corresponds to the time-lapse series shown in Fig. S1 and represents an independent biological replicate. Images were acquired every 10 s and displayed at 30 frames per second. The time stamp indicates minutes after the end of heat shock.

Video 8. **Second example of SUMO enrichment at a nuclear focus consistent with the preexisting nucleolar compartment prior to mitosis.** Time-lapse fluorescence video of hyphae co-expressing GFP-SUMO (green) and histone H1-RFP (magenta) during recovery from heat shock (42°C for 45 min). SUMO initially accumulates throughout the nucleus and then becomes selectively enriched at a single nuclear focus whose position and timing are consistent with the old nucleolar compartment. The SUMO-labeled remnant persists and can be observed in the cytoplasm following extrusion. This video represents an independent biological replicate. Images were acquired every 10 s and displayed at 30 frames per second. The time stamp indicates minutes after the end of heat shock.

Video 9. **Co-localization of Hsp70 and Hsp104 during nucleolar quality control.** Time-lapse fluorescence video of a hypha co-expressing Hsp70-RFP (magenta) and Hsp104-GFP (green) after heat shock (42°C for 45 min). Both chaperones enter a nucleus of the lower hypha on the right-hand side at ~104 min and become enriched at a nuclear focus corresponding to the preexisting nucleolus before remaining associated with the extruded remnant. Images were acquired every 10 s and displayed at 30 frames per second. The time stamp indicates minutes after the end of heat shock.

**Provided online is Table S1. Table S1 shows the primer list used in this study.**

