## [Peer Review File · The Journal of Cell Biology]

Stress-induced nucleolar rejuvenation via chaperone-mediated segregation in a filamentous fungus

Audra Rogers, Brenna Broussard, Rachel Taylor, and Martin Egan

Corresponding Author(s): Martin Egan, Merrimack College

Review Timeline:

Submission Date:	2025-08-11
Editorial Decision:	2025-09-18
Revision Received:	2025-12-18
Editorial Decision:	2026-01-31
Revision Received:	2026-02-04

Monitoring Editor: Ulrich Hartl

Scientific Editor: Dan Simon

Transaction Report:

DOI: <https://doi.org/10.1083/jcb.202508066>

September 18, 2025

Re: JCB manuscript #202508066

Martin Egan
Merrimack College

Dear Dr. Egan,

Thank you for submitting your manuscript entitled "Stress-induced nucleolar rejuvenation via chaperone-mediated segregation in a filamentous fungus." The manuscript was assessed by expert reviewers, whose comments are appended to this letter. We invite you to submit a revision if you can address the reviewers' key concerns, as outlined here.

You will see that the reviewers feel that your study presents new and interesting observations but a more thorough characterization of the novel nucleolar quality control pathway is necessary before publication. They ask for additional experiments to assess recruitment of other chaperones and to confirm that the old nucleolus contains damaged material which is extruded into the cytoplasm, both of which we agree would significantly strengthen the manuscript. Please also address the other comments asking for additional details, statistical info, and missing citations. Since this is a Report, investigating the mechanism in detail is not required.

GENERAL GUIDELINES:

Text limits: Character count for a Report is < 20,000, not including spaces. Count includes title page, abstract, introduction, the joint Results & Discussion, and acknowledgments. Count does not include materials and methods, figure legends, references, tables, or supplemental legends.

Figures: Reports may have up to 5 main text figures. To avoid delays in production, figures must be prepared according to the policies outlined in our Instructions to Authors, under Data Presentation, <https://jcb.rupress.org/site/misc/ifora.xhtml>. All figures in accepted manuscripts will be screened prior to publication.

*****IMPORTANT:** It is JCB policy that if requested, original data images must be made available. Failure to provide original images upon request will result in unavoidable delays in publication. Please ensure that you have access to all original microscopy and blot data images before submitting your revision. ***

Supplemental information: There are strict limits on the allowable amount of supplemental data. Reports may have up to 3 supplemental figures. Up to 10 supplemental videos or flash animations are allowed. A summary of all supplemental material should appear at the end of the Materials and methods section.

Please note that JCB now requires authors to submit Source Data used to generate figures containing gels and Western blots with all revised manuscripts. This Source Data consists of fully uncropped and unprocessed images for each gel/blot displayed in the main and supplemental figures. For assays performed using capillary electrophoresis and/or immunoassay-based detection, authors should instead provide the electropherogram graph(s) for each experiment, plotting fluorescence/chemiluminescence intensity vs. molecular weight/size. Please be sure to provide one Source Data file for each figure gels, blots, and/or capillary electrophoresis assays along with your revised manuscript files. File names for Source Data figures should be alphanumeric without any spaces or special characters (i.e., SourceDataF#, where F# refers to the associated main figure number or SourceDataFS# for those associated with Supplementary figures). For traditional gels and blots, the lanes of the gels/blots should be labeled as they are in the associated figure, the place where cropping was applied should be marked (with a box), and molecular weight/size standards should be labeled wherever possible. For capillary electrophoresis assays, each trace in the graph should be color-coded and labeled to indicate which protein, gene, or sample is being measured (please try to avoid red/green combinations to accommodate our color-blind readers).

The typical timeframe for revisions is three to four months. If you anticipate any difficulties in meeting this aforementioned revision time limit, please contact us and we can work with you to find an appropriate time frame for resubmission. Please note

that papers are generally considered through only one revision cycle, so any revised manuscript will likely be either accepted or rejected.

Thank you for this interesting contribution to Journal of Cell Biology. You can contact us at the journal office with any questions at cellbio@rockefeller.edu.

Sincerely,

Ulrich Hartl, PhD
Monitoring Editor
Journal of Cell Biology

Dan Simon, PhD
Scientific Editor
Journal of Cell Biology

Reviewer #1 (Comments to the Authors (Required)):

In this article, Rogers and colleagues studied the process of recovery from acute proteostatic stress of nucleoli in multinucleate syncytia in the filamentous fungus *Magnaporthe oryzae*. The authors describe a quality control pathway that ensures the inheritance of rejuvenated nucleolar material during nuclear division, while selectively eliminating the damaged one. The findings are interesting, although still in a preliminary form. The manuscript would benefit from a more accurate characterization of the newly identified process.

Major comments:

- the authors describe a novel nucleolar protein quality control pathway, yet they focus their attention only on Hsp104. The manuscript would benefit from a more accurate characterization of the components that are recruited along with Hsp104 at nucleoli during the discrimination process shown here. This could be done e.g. by co-staining using the old and new nucleoli with different antibodies for chaperones and co-chaperones or by testing the recruitment of other fluorescently labelled chaperones. Of note, the authors mention in the text that other chaperones and co-chaperones are known to bind to misfolded proteins and anchor them to deposition sites and they speculate that "these mechanisms could enable Hsp104 or its cofactors to selectively recognize the damaged nucleolus, distinguishing it from the newly assembled replacement". Experiments are needed here for better characterization.

- the authors suggest that the old nucleolus presumably harbors proteostatic damage from the heat shock. This should be experimentally addressed e.g. by co-staining the old and new nucleoli with antibodies for polyubiquitinated proteins or using other markers for misfolded proteins.

- the authors suggest that "the aberrant nucleolar material was being actively disaggregated or degraded (Fig. 4 C; Video 3)". Experiments should be carried out to discriminate these two fates, using e.g. proteasome and autophagy inhibitors and monitoring the fate of the aberrant nucleolar material.

- the authors speculate that the old nucleolus "acts as an extrudable phase-separated sink for misfolded nuclear proteins during recovery". There is no direct proof of material extrusion. Which would be the mechanism for extrusion of the aberrant nucleolar material? Could the authors inhibit specific enzymes required for the extrusion process to test their hypothesis?

Other comments:

- Please provide the appropriate references here: "The nucleolus is a membrane-less nuclear organelle that, in addition to ribosome biogenesis, contributes to the maintenance of nuclear proteostasis." and "Beyond its biosynthetic role, the nucleolus integrates growth and metabolic cues and undergoes rapid compositional and structural remodeling in response to cellular stress (Tiku and Antebi, 2018)." Examples include PMID: 31296649, PMID: 31271238, PMID: 31824572, PMID: 39690241.

- Please provide the appropriate references here: "Eukaryotic cells deploy spatial protein quality control systems that actively triage misfolded species to specific deposition sites according to their solubility, ubiquitination status, and refolding potential

(Miller et al., 2015; Kaganovich et al., 2008)." Examples include PMID: 37081164.

- "Most mechanistic insight of these deposition sites comes from budding yeast". Yet, later on in the text the authors refer to Frottin et al., 2019; Mediani et al., 2019; Latonen, 2019, who reported data from mammalian (human) cells. Please specify for clarity here.

"Thus, the nucleolus is both protective and a potential casualty of proteostatic collapse": I find this statement a bit strong and vague. Do we know whether a selective defect in the quality control function of nucleoli has an impact on the cytoplasmic protein quality control? Please rephrase to avoid misinterpretation.

- Please provide the appropriate references here: "Although the asymmetric inheritance of cytoplasmic protein aggregates to promote cellular rejuvenation is well established".

Reviewer #2 (Comments to the Authors (Required)):

Rogers et al. establish the conservation of nucleolar spatial protein quality control in a multinucleated organism *M. oryzae*. They describe an interesting phenotype of nucleolar budding, recruitment of Hsp104 to only the "old" nucleolus, and degradation of the "old" nucleolus. While extremely interesting, there are several statements that are not supported by the data and the paper is missing crucial experiments establishing how common this occurs and looking at additional chaperones known to be required for nucleolar protein quality control such as Hsp70.

Specific comments

- the text and figure legend do not mention the temperature or duration of the heat shock or the recovery

- the text states that the Hsp104-enriched structure was extruded into the cytoplasm, but Figure 1B lacks a visualization of the nucleus. How do you know that it is in the cytoplasm? The panels for 182 and 183 min have a dotted line that is presumably around the nucleus, but there is nothing to indicate the boundary of the nucleus when the extrusion is present.

- Fig 1c It is not clear why it takes so long from the apical nucleus to the tentative nucleolus? In panel B, there is only 1 min between the apical nucleus and the formation of the ring-like structure. Why would it take over 8 hours here?

- Similarly, in Fig 2B and C it takes 3.5 hours but typically recovery from heat shock is a very rapid phenomenon.

- The authors only tested Hsp104, and while very interesting, the paper would benefit from determining the involvement of other chaperones, at the very least Hsp70. Frottin et al showed a dependence on Hsp70, so it should be established if Hsp70 is required for the formation of this nucleolar bud. They also mention Hsp42, Sis1, and Btn2 but do not provide any data on the role of these sHsps in the process. They also do not mention Cur1 which is specifically shown to work with Sis1 to sequester nuclear material in Malinowska et al.

- The authors compare this extruded nucleolus to the INQ in yeast, but it is unclear if it can be formed by other stresses like DNA damage as was seen in Gallina et al. There is also no mention of Sontag et al 2023 where the INQ was extruded from the nucleus into the vacuole, or into the cytoplasm when the NVJ was disrupted.

- At the bottom of page 5, the authors state that the newly assembled structure is largely free from damage, but there is no data to support this conclusion. It doesn't recruit Hsp104, but that doesn't mean that it is not damaged and could be repaired by Hsp70 instead of degraded via Hsp104.

- the methods are light on details. For instance, there is no transformation method listed, the primer sequences are not included, there is no method for the recovery experiments, etc

- The figure legends could also use more details such as the temp and duration of heat shock, the recovery conditions, and the statistical test being used.

- Figure 3B is unclear what samples are being compared to generate the p values. Is it the intensity of green and purple in post compared to their intensity pre-conversion? Wouldn't that by necessity increase the intensity of red and decrease green? What is being tested in this experiment?

- it is also unclear how common this phenomenon is in a sample. Knowing how often it occurs in a data set will provide more impact and relevance to the study.

Reviewer #3 (Comments to the Authors (Required)):

The article by Rogers et al., describes some interesting observations of nucleolar chaperone segregation in post of stress-mediated nuclear division in the filamentous fungus *Magnaporthe oryzae*. The authors find that Hsp104 enters the nucleus of heat stress fungi during partial nuclear breakdown and localizes to the nucleolus. They used an elegant photoconvertible fluorescent protein strategy to label temporally old and new nucleoli, showing that Hsp104 was retained in the 'old' nucleolus, ultimately being extruded from the nucleus and disappearing. While the work presented is quite beautiful, limitations of the experimental system make it largely observational and lacking mechanistic insight. Most of the mechanism speculated is borrowed from other systems and not validated here.

1. Presumably the role of Hsp104 retention in old nucleoli is related to stabilizing heat denatured non-native proteins. The authors should show with model substrates, or thermo-labile endogenous proteins, that indeed non-native proteins are found in the 'old' nucleolus and interact with Hsp104 there.
2. The authors discussion outlines how Hsp42, Btn2, Sis1 or other factors could regulate this behaviour, but do they? Genetic experiments where one of these genes is deleted or other approaches to gene perturbation should be done so that a mechanistic model could begin to be assembled. Does small heat shock protein deletion/depletion alter the behaviour of Hsp104?
3. The authors use a catalytically inactive Hsp104 to show that the nucleolar structures are independent of ATPase was good. Is the HSP104DWB allele the only source of Hsp104 protein in the strain? Are there fitness consequences to losing the Hsp104 ATPase activity during heat stress? Some additional phenotypic characterization might help support interpretation of these data.
4. If the experimental system makes some of the suggestions challenging, are there other models of nucleolar segregation/aging/protein sequestration that could be incorporated to the study to test the mechanisms at play?
5. Do other proteotoxic stresses also cause the same behaviour or are the results restricted to heat shock? Does the timing of the heat shock or repeated heat shocks alter the results?

Dear Dr. Simon and Dr. Hartl,

We thank you and the reviewers for the thoughtful and constructive feedback on our manuscript. We have revised the paper extensively in response to the comments, including adding new experiments (SUMO enrichment, Hsp70 recruitment), clarifying methodological details, expanding figure legends, and refining the Discussion. We believe these changes substantially strengthen the manuscript and address all concerns raised.

Below, we provide a point-by-point response to the reviewers' comments in **magenta text**.

Response to Reviewers

Reviewer #1

Major comments:

- the authors describe a novel nucleolar protein quality control pathway, yet they focus their attention only on Hsp104. The manuscript would benefit from a more accurate characterization of the components that are recruited along with Hsp104 at nucleoli during the discrimination process shown here. This could be done e.g. by co-staining using the old and new nucleoli with different antibodies for chaperones and co-chaperones or by testing the recruitment of other fluorescently labelled chaperones. Of note, the authors mention in the text that other chaperones and co-chaperones are known to bind to misfolded proteins and anchor them to deposition sites and they speculate that "these mechanisms could enable Hsp104 or its cofactors to selectively recognize the damaged nucleolus, distinguishing it from the newly assembled replacement". Experiments are needed here for better characterization.

We thank the reviewer for this suggestion. To determine whether additional PQC components recognize the old nucleolar compartment, we generated a new strain co-expressing Hsp70–RFP and Hsp104–GFP. Hsp70 and Hsp104 co-localize on cytoplasmic aggregates after heat shock, enter the nucleus together at mitotic onset, and both become enriched at a single nuclear focus at the time and position expected for the pre-existing nucleolus. Both chaperones remain associated with the extruded remnant. These findings demonstrate that multiple chaperones, not only Hsp104, recognize the pre-existing nucleolar compartment during nucleolar quality control.

This new dataset is included as Video 9.

- the authors suggest that the old nucleolus presumably harbors proteostatic damage from the heat shock. This should be experimentally addressed e.g. by co-staining the old and new nucleoli with antibodies for polyubiquitinated proteins or using other markers for misfolded proteins.

We thank the reviewer for this important recommendation. We agree that directly assessing whether the pre-existing nucleolus contains stress-modified substrates is critical for interpreting its selective segregation. While polyubiquitin immunostaining is not readily compatible with the live-cell, time-resolved imaging conditions required for our system, we addressed the reviewer's request using a GFP–SUMO reporter, as SUMOylation is a well-established marker of stress-induced nuclear protein modification and often marks substrates destined for sequestration into nuclear PQC compartments.

Following heat shock, SUMO initially accumulates throughout the nucleus and then becomes selectively enriched at a single nuclear focus whose timing and position match the pre-existing ("old") nucleolar compartment identified in our Nop1 experiments. A SUMO-labeled remnant can also be observed after extrusion into the cytoplasm. Although SUMO marks a partially overlapping set of stress-modified proteins rather than ubiquitinated substrates specifically, these observations directly support the conclusion that the old nucleolar compartment contains stress-modified material consistent with proteostatic damage. These data are presented in Figure S1 and Videos 7–8, and are described in the Results

- the authors suggest that "the aberrant nucleolar material was being actively disaggregated or degraded (Fig. 4 C; Video 3)". Experiments should be carried out to discriminate these two fates, using e.g. proteasome and autophagy inhibitors and monitoring the fate of the aberrant nucleolar material.

We thank the reviewer for this helpful suggestion and agree that understanding the precise mechanism of remnant clearance is an important long-term goal. Our original language was intended to be descriptive rather than mechanistic, reflecting the clear observation that the Hsp104-marked remnant gradually diminishes and ultimately disappears after extrusion into the cytoplasm.

To avoid implying mechanistic resolution, we have revised the wording in the Results to state that the remnant diminishes "consistent with clearance by cytoplasmic quality control pathways," without specifying whether this involves disaggregation, degradation, or another process. Discriminating between these pathways in *M. oryzae*, for example by using proteasome or autophagy inhibition, would require substantial additional genetic and pharmacological optimization and falls outside the scope of the present work and the JCB Report format.

- the authors speculate that the old nucleolus "acts as an extrudable phase-separated sink for misfolded nuclear proteins during recovery". There is no direct proof of material extrusion. Which would be the mechanism for extrusion of the aberrant nucleolar material? Could the authors inhibit specific enzymes required for the extrusion process to test their hypothesis?

We thank the reviewer for raising this thoughtful point. We would like to clarify that in our manuscript the term "extruded" refers specifically to the physical displacement of the entire pre-existing nucleolar compartment out of the nucleus during mitosis, rather than extrusion of material from within the nucleolus. Our movies directly visualize this compartment being carried into the cytoplasm as a coherent remnant during semi-open mitosis in *M. oryzae*.

Thus, our use of "extruded" describes a morphological outcome of mitotic nuclear dynamics, rather than a mechanistically distinct export process acting on intranucleolar contents. We have not intended to imply that misfolded proteins are selectively expelled from the nucleolus by a specialized extrusion machinery.

Although understanding the forces and cellular components that govern this compartmental displacement is an important future direction, dissecting the molecular mechanism, for example through enzyme inhibition, is beyond the scope of the current study and the JCB Report format.

Other comments:

- Please provide the appropriate references here: "The nucleolus is a membrane-less nuclear

organelle that, in addition to ribosome biogenesis, contributes to the maintenance of nuclear proteostasis." and "Beyond its biosynthetic role, the nucleolus integrates growth and metabolic cues and undergoes rapid compositional and structural remodeling in response to cellular stress (Tiku and Antebi, 2018)." Examples include PMID: 31296649, PMID: 31271238, PMID: 31824572, PMID: 39690241.

We have now added several appropriate references to support the roles of the nucleolus in nuclear proteostasis and stress-induced remodeling. Specifically, we incorporated Frottin et al. (2019) and Mediani et al. (2019) to document nucleolar contributions to proteostasis, and we added Wang et al. (2019) and Brunello et al. (2025) to support rapid nucleolar compositional and structural changes during cellular stress. These citations have been included in the Introduction at the locations indicated by the reviewer.

- Please provide the appropriate references here: "Eukaryotic cells deploy spatial protein quality control systems that actively triage misfolded species to specific deposition sites according to their solubility, ubiquitination status, and refolding potential (Miller et al., 2015; Kaganovich et al., 2008)." Examples include PMID: 37081164.

Thank you for the suggestion. We have now added the recommended recent work on spatial protein quality control (Sontag et al., 2023; PMID 37081164) to the Introduction to further support this statement.

- "Most mechanistic insight of these deposition sites comes from budding yeast". Yet, later on in the text the authors refer to Frottin et al., 2019; Mediani et al., 2019; Latonen, 2019, who reported data from mammalian (human) cells. Please specify for clarity here.

We thank the reviewer for this helpful observation. To clarify the distinction between yeast-based mechanistic insight and related findings in mammalian systems, we have revised the relevant sentence in the Introduction. It now explicitly states that most mechanistic understanding derives from budding yeast, although comparable sequestration mechanisms have also been reported in mammalian cells (Frottin et al., 2019; Mediani et al., 2019; Latonen, 2019). This revision should resolve the ambiguity noted by the reviewer.

"Thus, the nucleolus is both protective and a potential casualty of proteostatic collapse": I find this statement a bit strong and vague. Do we know whether a selective defect in the quality control function of nucleoli has an impact on the cytoplasmic protein quality control? Please rephrase to avoid misinterpretation.

To avoid overstating the role of the nucleolus and to prevent any unintended mechanistic implications, we have revised the sentence in the Introduction. It now reads: "Thus, while the nucleolus contributes to maintaining nuclear proteostasis, it can also undergo stress-induced alterations, and how its normal architecture is re-established during recovery remains unclear."

This revised phrasing softens the original claim and removes any suggestion of a direct impact on cytoplasmic proteostasis.

- Please provide the appropriate references here: "Although the asymmetric inheritance of cytoplasmic protein aggregates to promote cellular rejuvenation is well established".

We have now added several well-established studies documenting asymmetric inheritance of cytoplasmic protein aggregates in budding yeast, including Aguilaniu et al. (2003), Kaganovich

et al. (2008), and Spokoini et al. (2012). These citations have been incorporated into the Introduction to support this statement.

Reviewer #2 (Comments to the Authors (Required)):

Specific comments

- the text and figure legend do not mention the temperature or duration of the heat shock or the recovery

We thank the reviewer for pointing this out. We have now added the heat-shock conditions (42°C for 45 min) and recovery details throughout the manuscript, including in the main text and in all relevant figure and video legends (Figs. 1–4; Videos 1–9).

- the text states that the Hsp104-enriched structure was extruded into the cytoplasm, but Figure 1B lacks a visualization of the nucleus. How do you know that it is in the cytoplasm? The panels for 182 and 183 min have a dotted line that is presumably around the nucleus, but there is nothing to indicate the boundary of the nucleus when the extrusion is present.

We thank the reviewer for this important clarification. We agree that Figure 1B alone does not provide sufficient evidence to support a definitive claim of cytoplasmic extrusion, as the nuclear boundary is not directly visualized in this specific hypha. In response, we have removed the explicit reference to cytoplasmic extrusion from the Fig. 1 section and now describe only the observed disassembly of the Hsp104-enriched structure during mitotic progression. At the same time, we note that subsequent figures and movies in the manuscript unambiguously demonstrate that the pre-existing nucleolar remnant resides outside of the daughter nuclei. For example, in Figure 2B and Figure 3F, as well as in accompanying videos, the old nucleolar compartment is clearly positioned between or outside separated daughter nuclei, where both nuclear and nucleolar markers are present simultaneously. These datasets provide strong support for the conclusion that the old nucleolar remnant is extruded from the nucleus and cleared in the cytoplasm.

We appreciate the reviewer's attention to this point and have clarified the text accordingly.

- Fig 1c It is not clear why it takes so long from the apical nucleus to the tentative nucleolus? In panel B, there is only 1 min between the apical nucleus and the formation of the ring-like structure. Why would it take over 8 hours here?

We thank the reviewer for pointing out this source of confusion. The apparent difference in timing between Fig. 1B and Fig. 1C arises from two factors.

First, Fig. 1C shows the behavior of the ATPase-deficient Hsp104^{DWB} mutant, which displays markedly altered aggregation and recovery kinetics compared to the wild-type Hsp104 shown in Fig. 1B. In the Hsp104^{DWB} background, heat-induced aggregates persist much longer, and recovery to the point of mitotic re-entry is substantially delayed, which accounts for the extended time before nucleolar localization events become apparent.

Second, the issue regarding the apical nucleus to tentative nucleolus timing reflects a labeling/visualization issue rather than a biological interval. In Fig. 1C we chose to begin the time series at a moment when the cytoplasmic aggregates were not yet visible (0 min) to provide a clear baseline for the Hsp104^{DWB} phenotype. By contrast, Fig. 1B does not include an

equivalent early timepoint. This difference in presentation inadvertently contributed to the perceived discrepancy.

To clarify the temporal dynamics in the Hsp104^{DWB} strain, we now include a full-length time-lapse movie corresponding to Fig. 1C (Video 2). This movie displays the entire recovery period and progression to mitosis, making the altered kinetics of the Hsp104^{DWB} mutant clear.

- Similarly, in Fig 2B and C it takes 3.5 hours but typically recovery from heat shock is a very rapid phenomenon.

We thank the reviewer for this observation. In contrast to single-celled systems where heat-shock recovery can be rapid, polarized, multinucleate syncytia such as *M. oryzae* hyphae exhibit more variable and often prolonged recovery periods. Acute heat shock causes a sustained arrest of both polarized tip growth and nuclear movement in this system (Fig. 1A), and nuclei re-enter the cell cycle only after this extended recovery phase. The ~3.5-hour interval in Fig. 2B–C therefore reflects this syncytial recovery behavior, whereas the nucleolar remodeling itself occurs immediately prior to the first post-stress mitosis.

- The authors only tested Hsp104, and while very interesting, the paper would benefit from determining the involvement of other chaperones, at the very least Hsp70. Frottin et al showed a dependence on Hsp70, so it should be established if Hsp70 is required for the formation of this nucleolar bud. They also mention Hsp42, Sis1, and Btn2 but do not provide any data on the role of these sHsps in the process. They also do not mention Cur1 which is specifically shown to work with Sis1 to sequester nuclear material in Malinovska et al.

We thank the reviewer for this valuable suggestion. In response, we generated a new strain co-expressing Hsp70-RFP and Hsp104-GFP to examine whether additional chaperones participate in recognizing the pre-existing nucleolar compartment. As shown in Video 9, Hsp70 and Hsp104 co-localize on cytoplasmic stress-induced aggregates, enter the nucleus together at mitotic onset, and become enriched at a nuclear focus consistent with the pre-existing nucleolus, where they remain associated with the extruded remnant. These observations demonstrate that Hsp70 participates in this nucleolar quality control pathway and strengthen the link to chaperone-mediated segregation mechanisms described in other systems.

We also attempted to generate a Nop1-Hsp70 strain to directly visualize Hsp70 localization relative to the old nucleolar compartment, analogous to our Hsp104 experiments, but encountered technical challenges in constructing this strain in *M. oryzae*.

We agree that factors such as Hsp42, Sis1, Btn2, and Cur1 play important roles in spatial protein quality control in budding yeast. However, systematically dissecting the contributions of these pathways in *M. oryzae* would require extensive genetic perturbation and falls beyond the scope of the present work and the JCB Report format.

- The authors compare this extruded nucleolus to the INQ in yeast, but it is unclear if it can be formed by other stresses like DNA damage as was seen in Gallina et al. There is also no mention of Sontag et al 2023 where the INQ was extruded from the nucleus into the vacuole, or into the cytoplasm when the NVJ was disrupted.

We thank the reviewer for this helpful point. Our comparison to the INQ was intended purely as a conceptual parallel, not as a claim of mechanistic equivalence or formation under the same stresses. We have only examined nucleolar remodeling in response to acute heat shock, and we do not yet know whether other stresses trigger similar behavior in *M. oryzae*. To provide clearer context, we now note in the Discussion that the INQ in yeast can be evicted from the

nucleus under certain quality-control conditions (Sontag et al., 2023). This revision clarifies the analogy without implying that the underlying pathways are shared.

- At the bottom of page 5, the authors state that the newly assembled structure is largely free from damage, but there is no data to support this conclusion. It doesn't recruit Hsp104, but that doesn't mean that it is not damaged and could be repaired by Hsp70 instead of degraded via Hsp104.

We thank the reviewer for this thoughtful point. We agree that our original phrasing could be interpreted as stronger than our data support. To avoid overclaiming, we have revised the wording to clarify that we are speculating that the newly assembled nucleolar compartment is largely free of stress-induced damage, based on its distinct behavior relative to the pre-existing compartment. The revised sentence now reads:

“One is a newly assembled structure that we speculate is largely free of damage and destined for inheritance.”

This phrasing more accurately reflects the observational nature of our study.

- the methods are light on details. For instance, there is no transformation method listed, the primer sequences are not included, there is no method for the recovery experiments, etc

We thank the reviewer for noting these omissions. We have substantially expanded the Materials and Methods section to include all missing details.

- The figure legends could also use more details such as the temp and duration of heat shock, the recovery conditions, and the statistical test being used.

We have now expanded all relevant figure legends to include the temperature and duration of the heat shock (42 °C for 45 min), the recovery conditions, and the specific statistical tests used in each analysis.

- Figure 3B is unclear what samples are being compared to generate the p values. Is it the intensity of green and purple in post compared to their intensity pre-conversion? Wouldn't that by necessity increase the intensity of red and decrease green? What is being tested in this experiment?

In Fig. 3B, the paired measurements compare pre- and post-photoconversion fluorescence intensities within the same nucleolus for each channel independently (green = non-converted mEos, magenta = photoconverted mEos). The purpose of this analysis is not to test biological differences between compartments but simply to verify the efficiency and consistency of the photoconversion step across nuclei. As expected, photoconversion increases magenta signal and decreases green signal within each nucleolus. The statistical test (paired t-test) confirms that the conversion behaved consistently across samples. We have clarified this in the respective figure legend.

- it is also unclear how common this phenomenon is in a sample. Knowing how often it occurs in a data set will provide more impact and relevance to the study.

We have now quantified the frequency of nucleolar remodeling across multiple biological replicates. Among 35 apical nuclei examined across 10 independent experiments, 23 nuclei

(~66%) showed nucleolar budding, selective Hsp104 recruitment, and extrusion of the pre-existing nucleolar remnant during the first mitosis after heat shock. This penetrance value is now reported in the Results to clarify how commonly the phenomenon occurs.

Reviewer #3 (Comments to the Authors (Required)):

1. Presumably the role of Hsp104 retention in old nucleoli is related to stabilizing heat denatured non-native proteins. The authors should show with model substrates, or thermo-labile endogenous proteins, that indeed non-native proteins are found in the 'old' nucleolus and interact with Hsp104 there.

We thank the reviewer for this insightful suggestion. We agree that directly demonstrating the presence of non-native or misfolded proteins within the pre-existing nucleolar compartment would further illuminate the underlying mechanism. While introducing model substrates or thermo-labile reporters into *M. oryzae* presents substantial technical challenges, we have addressed this question using an alternative approach that is compatible with live-cell imaging in this system.

Specifically, we generated a GFP-SUMO reporter strain to monitor the accumulation of stress-modified nuclear proteins. SUMOylation is a well-established marker of proteins that have undergone stress-induced modification and are targeted to nuclear quality-control pathways. Following heat shock, we observed a selective enrichment of SUMO at a single nuclear focus whose position and timing are consistent with the pre-existing ("old") nucleolar compartment. A SUMO-positive remnant also persists transiently after extrusion into the cytoplasm. These findings indicate that the old nucleolar compartment contains SUMO-modified substrates, consistent with the presence of stress-altered proteins.

These new data are presented in Figure S1 and Videos 7-8, and are described in the Results section. Although we have not directly tested specific thermo-labile substrates, we believe that the selective SUMO enrichment provides reasonable evidence that the pre-existing nucleolus is enriched in stress-modified or non-native proteins. Further defining the precise protein species present in this remnant represents an interesting direction for future work.

2. The authors discussion outlines how Hsp42, Btn2, Sis1 or other factors could regulate this behaviour, but do they? Genetic experiments where one of these genes is deleted or other approaches to gene perturbation should be done so that a mechanistic model could begin to be assembled. Does small heat shock protein deletion/depletion alter the behaviour of Hsp104?

We thank the reviewer for this thoughtful suggestion. We agree that dissecting the roles of factors such as Hsp42, Btn2, Sis1, or Cur1 would be informative for understanding how misfolded proteins are triaged toward the old nucleolar compartment. In keeping with this recommendation, we performed an additional experiment to assess whether other chaperones participate in this pathway. As described in the revised Results, Hsp70-RFP and Hsp104-GFP co-localize on cytoplasmic aggregates, enter the nucleus together, and become enriched at the pre-existing nucleolar compartment (Video 9). This finding strengthens the connection between nucleolar remodeling and established PQC pathways.

However, systematically deleting or perturbing multiple PQC components in *M. oryzae* represents a substantial undertaking and is beyond the scope of the present study.

3. The authors use a catalytically inactive Hsp104 to show that the nucleolar structures are independent of ATPase was good. Is the HSP104DWB allele the only source of Hsp104 protein in the strain? Are there fitness consequences to losing the Hsp104 ATPase activity during heat

stress? Some additional phenotypic characterization might help support interpretation of these data.

We thank the reviewer for raising these important points. In the Hsp104^{DWB} strain used in this study, the Hsp104^{DWB} allele is the sole source of Hsp104, and no wild-type copy of the gene is present. Although we did not perform additional phenotypic characterization of this allele in the current work, the aggregation dynamics and functional consequences of expressing Hsp104^{DWB} in *M. oryzae* were described in detail in our previous study (Rogers and Egan, 2020; PMID: 32816646). In that work, the mutant behaved as expected for a catalytically inactive disaggregase, displaying persistent aggregates and altered recovery kinetics under stress, consistent with what we observe here.

Because our use of Hsp104^{DWB} in this manuscript was limited to determining whether nucleolar remodeling requires Hsp104's ATPase activity, and because the mutant produced a clear and interpretable phenotype in this context, we did not expand the phenotypic analysis further.

4. If the experimental system makes some of the suggestions challenging, are there other models of nucleolar segregation/aging/protein sequestration that could be incorporated to the study to test the mechanisms at play?

We thank the reviewer for this thoughtful suggestion. In this study, our goal was to define and characterize a stress-induced nucleolar quality-control pathway within the native context of a polarized, multinucleate fungal syncytium, as this system presents unique spatial and cell-cycle constraints that cannot be readily recapitulated in other models. Incorporating additional organisms or model systems to test mechanisms would require substantial new experimental development and lies beyond the scope of this study.

Instead, to strengthen the mechanistic grounding within *M. oryzae*, we introduced several new approaches compatible with this system, including SUMO labeling to detect stress-modified substrates (Fig. S1; Videos 7–8) and Hsp70–RFP/Hsp104–GFP imaging to assess participation of additional chaperones (Video 9). These additions reinforce the conclusion that the pre-existing nucleolar compartment is selectively recognized by nuclear PQC machinery.

We agree that comparative analyses across systems could provide valuable mechanistic insight, and we view this as an exciting direction for future work.

5. Do other proteotoxic stresses also cause the same behaviour or are the results restricted to heat shock? Does the timing of the heat shock or repeated heat shocks alter the results?

We thank the reviewer for this thoughtful question. In the present study, we focused specifically on acute heat shock, as this treatment reliably induces proteotoxic stress in *M. oryzae* and produces a robust and reproducible nucleolar remodeling response across biological replicates. We have not yet tested whether other stresses (such as oxidative or genotoxic insults) trigger similar behavior, nor have we examined alternative heat-shock timing paradigms or repeated stress cycles. These represent interesting avenues for future work, but addressing them would require substantial additional experimentation and lies beyond the scope of this report.

January 31, 2026

RE: JCB Manuscript #202508066R

Martin Egan
Merrimack College

Dear Dr. Egan,

Thank you for submitting your revised manuscript entitled "Stress-induced nucleolar rejuvenation via chaperone-mediated segregation in a filamentous fungus." The manuscript was re-assessed by two of the original reviewers; the third reviewer was not available. We would be happy to publish your paper in JCB pending the minor textual changes recommended by the reviewers as well as final revisions necessary to meet our formatting guidelines (see details below).

A. MANUSCRIPT ORGANIZATION AND FORMATTING:

1) Text limits: Character count for Reports is < 20,000, not including spaces. Count includes title page, abstract, introduction, results & discussion, and acknowledgments. Count does not include materials and methods, figure legends, references, tables, or supplemental legends.

2) Figure formatting: Reports may have up to 5 main text figures. Scale bars must be present on all microscopy images, including inset magnifications. Also, please avoid pairing red and green for images and graphs to ensure legibility for color-blind readers. If red and green are paired for images, please ensure that the particular red and green hues used in micrographs are distinctive with any of the colorblind types. If not, please modify colors accordingly or provide separate images of the individual channels.

3) Statistical analysis: Error bars on graphic representations of numerical data must be clearly described in the figure legend. The number of independent data points (n) represented in a graph must be indicated in the legend. Please indicate whether 'n' refers to technical or biological replicates (i.e. number of analyzed cells, samples or animals, number of independent experiments). If independent experiments with multiple biological replicates have been performed, we recommend using distribution-reproducibility SuperPlots (please see Lord et al., JCB 2020) to better display the distribution of the entire dataset, and report statistics (such as means, error bars, and P values) that address the reproducibility of the findings.

Statistical methods should be explained in full in the materials and methods. For figures presenting pooled data the statistical measure should be defined in the figure legends. Please also be sure to indicate the statistical tests used in each of your experiments (both in the figure legend itself and in a separate methods section) as well as the parameters of the test (for example, if you ran a t-test, please indicate if it was one- or two-sided, etc.). Also, if you used parametric tests, please indicate if the data distribution was tested for normality (and if so, how). If not, you must state something to the effect that "Data distribution was assumed to be normal but this was not formally tested."

4) Materials and methods: Should be comprehensive and not simply reference a previous publication for details on how an experiment was performed. Please provide full descriptions (at least in brief) in the text for readers who may not have access to referenced manuscripts. The text should not refer to methods "...as previously described."

5) For all cell lines, vectors, strains, constructs/cDNAs, etc. - all genetic material: please include database / vendor ID (e.g. Addgene, ATCC, etc.) or if unavailable, please briefly describe their basic genetic features, even if described in other published work or gifted to you by other investigators (and provide references where appropriate). Please be sure to provide the sequences for all of your oligos: primers, si/shRNA, RNAi, gRNAs, etc. in the materials and methods. You must also indicate in the methods the source, species, and catalog numbers/vendor identifiers (where appropriate) for all of your antibodies, including secondary. If antibodies are not commercial, please add a reference citation if possible.

6) Microscope image acquisition: The following information must be provided about the acquisition and processing of images:

- a. Make and model of microscope
- b. Type, magnification, and numerical aperture of the objective lenses
- c. Temperature
- d. Imaging medium
- e. Fluorochromes

f. Camera make and model

g. Acquisition software

h. Any software used for image processing subsequent to data acquisition. Please include details and types of operations involved (e.g., type of deconvolution, 3D reconstitutions, surface or volume rendering, gamma adjustments, etc.).

7) References: There is no limit to the number of references cited in a manuscript. References should be cited parenthetically in the text by author and year of publication. Abbreviate the names of journals according to PubMed.

8) Supplemental materials: Reports may have up to 3 supplemental figures and 10 videos. Please also note that tables, like figures, should be provided as individual, editable files. A summary of all supplemental material should appear at the end of the Materials and methods section. Please include one brief sentence per item.

9) Video legends: Each video should have a separate legend describing what is being shown, the cell type or tissue being viewed (including relevant cell treatments, concentration and duration, or transfection), the imaging method (e.g., time-lapse epifluorescence microscopy), what each color represents, how often frames were collected, the frames/second display rate, and the number of any figure that has related video stills or images.

10) eTOC summary: A ~40-50 word summary that describes the context and significance of the findings for a general readership should be included on the title page. The statement should be written in the present tense and refer to the work in the third person. It should begin with "First author name(s) et al..." to match our preferred style.

11) Conflict of interest statement: JCB requires inclusion of a statement in the acknowledgements regarding competing financial interests. If no competing financial interests exist, please include the following statement: "The authors declare no competing financial interests." If competing interests are declared, please follow your statement of these competing interests with the following statement: "The authors declare no further competing financial interests."

12) A separate author contribution section is required following the Acknowledgments in all research manuscripts. All authors should be mentioned and designated by their first and middle initials and full surnames. We encourage use of the CRediT nomenclature (<https://casrai.org/credit/>).

13) ORCID IDs: ORCID IDs are unique identifiers allowing researchers to create a record of their various scholarly contributions in a single place. Please note that ORCID IDs are required for all authors. At resubmission of your final files, please be sure to provide your ORCID ID and those of all co-authors.

14) Journal of Cell Biology now requires a data availability statement for all research article submissions. These statements will be published in the article directly above the Acknowledgments. The statement should address all data underlying the research presented in the manuscript. Please visit the JCB instructions for authors for guidelines and examples of statements at (<https://rupress.org/jcb/pages/editorial-policies#data-availability-statement>).

B. FINAL FILES:

-- An editable version of the final text (.DOC or .DOCX) is needed for copyediting (no PDFs). Please highlight all changes in the text.

Additionally, JCB encourages authors to submit a short video summary of their work. These videos are intended to convey the main messages of the study to a non-specialist, scientific audience. Think of them as an extended version of your abstract, or a

short poster presentation. We encourage first authors to present the results to increase their visibility. The videos will be shared on social media to promote your work. For more detailed guidelines and tips on preparing your video, please visit <https://rupress.org/jcb/pages/submission-guidelines#videoSummaries>.

Thank you for your attention to these final processing requirements. Please revise and format the manuscript and upload materials within 7-14 days. If you need an extension for whatever reason, please let us know and we can work with you to determine a suitable revision period.

Thank you for this interesting contribution, we look forward to publishing your paper in Journal of Cell Biology.

Sincerely,

Ulrich Hartl, PhD
Monitoring Editor
Journal of Cell Biology

Dan Simon, PhD
Scientific Editor
Journal of Cell Biology

Reviewer #1 (Comments to the Authors (Required)):

The authors have strengthened the manuscript by adding new experiments and revising several sections of the text. However, some conclusions still require careful rephrasing to avoid overinterpretation.

I appreciate the experiments performed using a GFP-SUMO reporter, since SUMO conjugation levels are known to increase upon exposure to several stress conditions in yeast cells. The manuscript would benefit from a brief sentence summarizing the established roles of SUMO (Smt3) in yeast physiology and stress responses, supported by at least one relevant reference.

The following sentence also would benefit of appropriate references: "These observations suggest that multiple PQC factors, including Hsp70 and Hsp104, recognize the old nucleolar compartment prior to its segregation, consistent with a conserved role for chaperone networks in managing stress-modified nuclear material."

The following sentences should be accordingly revised:

- 1) "the extruded nuclear remnant contains SUMO-modified material and support a model in which this compartment is selectively marked for downstream clearance during recovery";
- 2) "Whether the extruded nucleolus is truly damaged or simply serves as a sequestration site remains unresolved";
- 3) "The other is an older remnant that accumulates SUMO-modified material, recruits the chaperones Hsp70 and Hsp104, and is subsequently extruded into the cytoplasm. We propose that this selective segregation limits transmission of damaged nucleolar components to daughter nuclei and safeguards nuclear proteostasis during recovery".

Given that the authors have not demonstrated that the extruded material undergoes clearance-and indeed suggest that this process may function as a sequestration mechanism for asymmetric damage segregation-I recommend rephrasing the first sentence to refer to sequestration rather than clearance, ensuring alignment with points (2) and (3).

Figure 1B still lacks a visualization of the nucleus; overall the nuclear boundary is not directly visualized. Although the authors state that Figures 2 and 3 support the conclusion that the old nucleolar remnant is extruded from the nucleus, the claim of cytoplasmic extrusion for clearance should be revised throughout the manuscript.

Methods: Please specify in the methods that GFP-SUMO corresponds to yeast SUMO (Smt3).

Reviewer #3 (Comments to the Authors (Required)):

The authors have provided a thorough and thoughtful revision to the manuscript. While they have not been able to address everything that was asked, in large part due to the experimental system, the new data on Hsp70 and SUMO localization is important.

If additional revisions are made, they should think carefully about the correspondence of their observations with INQ (yeast) and with human nucleoli. Surely there is some evolutionarily conserved pathway underlying all of these observations (the new data with SUMO ties this even more strongly to INQ in my opinion where SUMO is also implicated). I think it would be fair for the

authors to speculate more.

Dear Dr. Hartl and Dr. Simon,

We thank you for the opportunity to submit a revised version of our manuscript following acceptance in principle. In response to the remaining reviewer comments, we have made the following targeted revisions to the text.

We added a brief contextual sentence summarizing established roles of SUMO (Smt3) in nuclear physiology and stress responses, supported by appropriate references, at the point where SUMO dynamics are introduced. We also added a reference supporting the statement that multiple protein quality control factors, including Hsp70 and Hsp104, participate in the recognition and handling of stress-modified nuclear material.

To avoid overinterpretation, we revised several sentences in the Results and Discussion to remove language implying demonstrated cytoplasmic clearance of the extruded nucleolar material. The text now consistently frames this compartment as a sequestered remnant that may undergo downstream processing, while explicitly noting that its ultimate fate cannot be resolved from the present data.

In response to reviewer feedback, we also expanded the Discussion to more explicitly place our observations in the context of intranuclear quality control pathways described in yeast and stress-responsive nucleolar remodeling in mammalian cells, while keeping these connections appropriately speculative.

In addition, the Methods section has been updated to specify that the GFP-SUMO reporter corresponds to yeast SUMO (Smt3). We also made minor editorial and methodological revisions throughout the manuscript to improve clarity and ensure compliance with JCB formatting and reporting standards.

All changes in this revision are highlighted in the marked manuscript PDF provided.

With thanks,

Martin